

# Causes of the exceptionally high number of fatalities in the Ahr valley, Germany, during the 2021 flood

Belinda Rhein[1,2], Heidi Kreibich[1]

1 GFZ German Research Centre for Geosciences, Section Hydrology, Potsdam, Germany

2 Humboldt-Universität zu Berlin, Geography Department, Berlin, Germany

Correspondence to:

Belinda Rhein, belinda.rhein@hu-berlin.de

Heidi Kreibich, heidi.kreibich@gfz-potsdam.de

**Key Points:**

With 190 fatalities, 134 of them in the Ahr Valley, the 2021 event was the deadliest flood in recent German history.

Many people died on the ground floor (37%) or outside on the street (18%), elderly over the age of 60 were particularly vulnerable (78%).

Before extreme flash floods, warnings must make it clear that saving lives takes priority and evacuations must be carried out in good time, paying particular attention to the elderly.

**Abstract**

Over the last 40 years (1980-2020), 159 people have died in inland floods in Germany. The flood of 2021 caused 190 flood fatalities in Germany, 134 of them in the Ahr valley. We investigate what made this event so 'deadly' in order to help improve flood risk management and prevent future fatalities. A comprehensive analysis of the factors influencing the occurrence of fatalities is carried out on the basis of the death investigation files of the public prosecutor's office. This unprecedented flash flood was characterised by high water levels and high flow velocities. The extent of inundation in 2021 far exceeded the official hazard map for the extreme flood scenario. Additionally, early warning and evacuation were inadequate so that many people were surprised by the flash flood. 75% of the fatalities occurred outside of the mapped hazard zones. Particularly dangerous places were campsites, cellars and basement flats, but many people died on the ground floor (37%) or outside on the street (18%). The elderly above 60 years of age (78%) and those with mobility or cognitive impairments (16%) were particularly vulnerable. No gender-specific differences in vulnerability were observed. Public understanding of the particular danger posed by flash floods must be improved, as must the development and presentation of worst-case scenarios in hazard maps. Additionally, impact forecasting can significantly improve emergency management of such unprecedented floods. Specific recommendations are that in the event of such extreme flash floods, the warning messages must clearly communicate that saving human lives must be the priority, i.e., those at risk should move to safe places, e.g., to the upper floors. Evacuations must be initiated in good time, especially where flooding of the ground floor with high water levels is to be expected, paying particular attention to the safety of the elderly and people with limited mobility.



**Plain Language Summary**

The flood of 2021 killed 190 people in Germany, 134 of them in the Ahr valley, making it the deadliest flood in recent German history. The flash flood was extraordinarily extreme in terms of water levels, flow velocities and flood extent. In addition, early warning and evacuation were inadequate. Many people died on the ground floor or in the street, places that are not commonly perceived as particularly dangerous. Older people over the age of 60 were particularly vulnerable as well as people with mobility or cognitive impairments. In the event of such extreme flash floods, warnings should clearly state that those at risk should move to safe places, such as upper floors, and not try to save their belongings, especially not from the basement. Evacuations must be initiated in good time, with particular attention to the safety of the elderly.

## 1. Introduction

The flood of July 2021 can be described as an unprecedented flash flood (Kreibich et al., 2022) with a particularly high number of deaths, especially with 134 fatalities in the Ahr valley in Germany (Koks et al., 2021). This number of fatalities is exceptionally high for Germany, with the most recent deadliest floods causing 11 deaths in 2016, 14 in 2013, and 21 in 2002 (Papagiannaki et al., 2022). In the context of inland floods in Germany, fewer people died in the last 40 years between 1980 and 2020, with 159 victims (Petrucci et al., 2022), than in this single flood event in July 2021 with 190 victims (Figure 1).

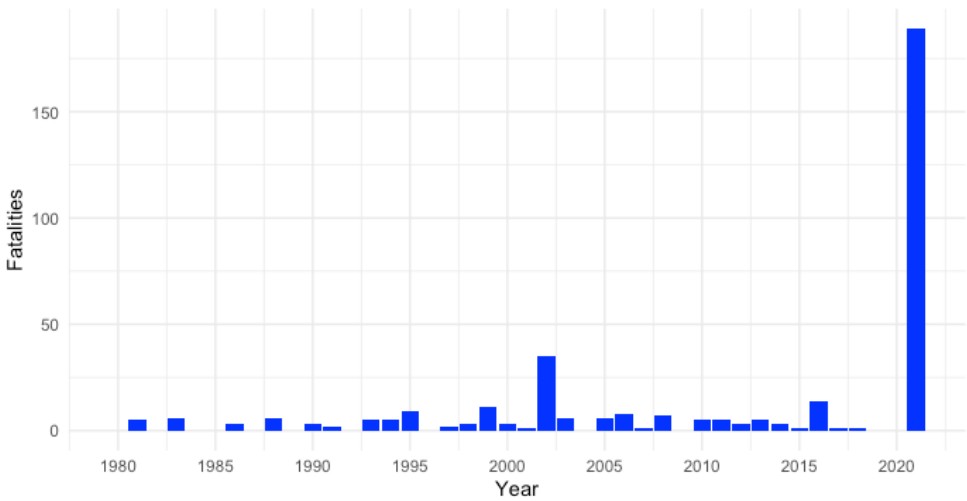

Figure 1: Fatalities in inland floods in Germany (1980-2021) according to Petrucci et al. (2022) and 2021 event added.

Detailed knowledge about the causes of flood fatalities is scarce, due to a lack of data. Most often only the number of fatalities is available, sometimes including information on gender and age (Kellar & Schmidlin, 2012; Paul & Mahmood, 2016; Pereira et al., 2017; Sharif et al., 2012). Exceptions include Thieken et al. (2023a) analyzing the flood fatalities in North Rhine-Westphalia for the same flood in great detail. They found that the elderly were particularly



vulnerable, with lack of warning and lack of flood awareness as main causes of flood fatalities.
Ahmed et al. (2020), were able to analyze vehicle-related flood fatalities in Australia between
2001 and 2017 accessing police statements and forensic reports. They identified middle-aged
and elderly males as the most common fatalities as drivers, while young women and children
were most vulnerable as passengers (Ahmed et al., 2020). Diakakis and Deligiannakis (2017)
developed a detailed database of more than 150 flood fatalities that occurred in Greece between
1970 and 2010 and found that accidents mostly occurred at night outdoors in rural areas, with
men and elderly as most vulnerable, and vehicle-related fatalities as the most common.
Hydrologically, the 2021 flood was extreme in terms of the rapid onset of flooding, high flow
velocities, high water depths and large inundation extent. Between the 12$^{th}$ and the 19$^{th}$ of July
2021, the low-pressure system "Bernd" resulted in extreme rainfall of more than 150 mm in 72
hours (Mohr et al., 2022). The high rainfall on already saturated soils led to surface runoff,
especially along the narrow valley of the river Ahr (Kron et al., 2022). Water levels started
rising between 8 and 10 am on the 14$^{th}$ of July and peaked at around 8 to 10 pm on the same
day at the Ahr gauge Müsch upstream (Figure 2). Further downstream gauges showed peak
water levels during the night and in the early morning of the 15$^{th}$ of July, namely at the gauge
Altenahr between 0 and 1 am (Mohr et al., 2022).

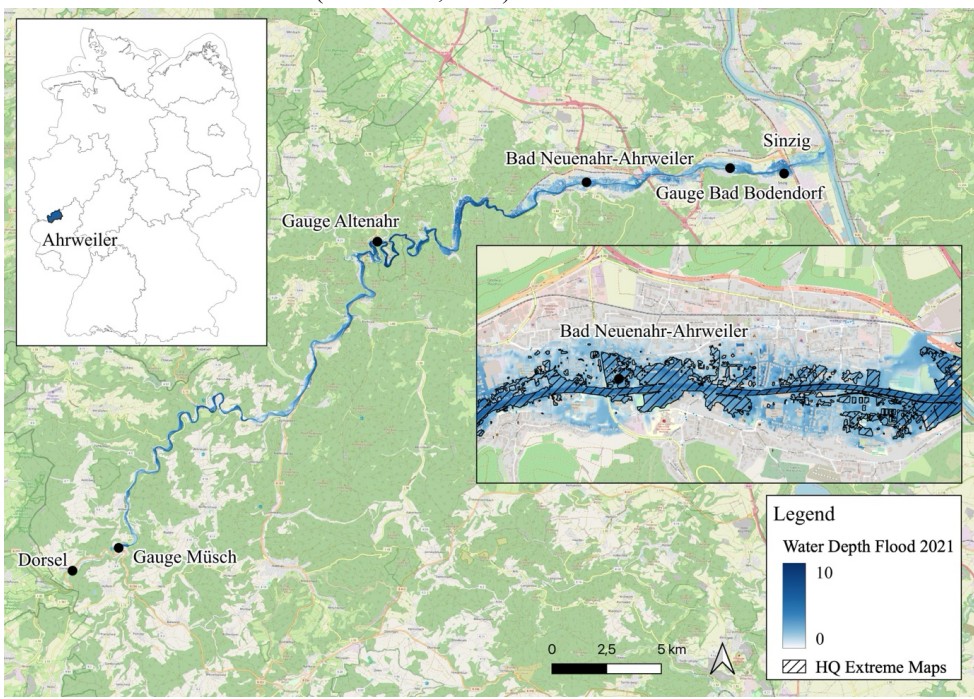

Figure 2: Study Site along the Ahr with gauges (Landesamt für Umwelt Rheinland-Pfalz,
2016), water depths of the Flood 2021 (Apel et al., 2022) and the HQ Extreme Maps of 2018
for Bad Neuenahr-Ahrweiler (Ministerium für Klimaschutz, Umwelt, Energie und Mobilität
Rheinland-Pfalz, 2018).
© OpenStreetMap contributors 2024. Distributed under the Open Data Commons Open
Database License (ODbL) v1.0.





The official hazard map available in July 2021 for the extreme flood scenario for the Ahr Valley
significantly underestimated the inundation area of the 2021 flood (Figure 2). The hazard map
was calculated on the basis of the flow records from 1947, when the continuous recording of
water levels had begun but which did not include such extreme events (Kron et al., 2022).
There had been warnings issued by the German meteorological service starting on the 11th of
July 2021 for the potentially flood-triggering low-pressure system. The forecast that was
published 24 hours before the event suggested a maximum water depth of 5.74 meters at the
gauge in Altenahr, while reconstructions of the event show that the peak water levels were at
about 10.2 meters (Apel et al., 2022). The district of Ahrweiler released a flood warning on the
14th of July 2021 in the early afternoon (Thieken et al., 2023b). At 11.09 pm on the 14th of July,
the state of emergency was declared in the municipality of Altenahr and residents 50 metres on
either side of the Ahr were asked to evacuate, although the flood water was already dangerously
high at this time (Thieken et al., 2023b). The local authorities apparently underestimated the
flood and the official warnings did not convey the extreme severity of the impending flood.
Thieken et al. (2023b) showed that 29% of those affected by the 2021 flood in Rhineland-
Palatinate had not received any warning.
The 2021 flood was also extreme in terms of its consequences with 190 fatalities and economic
losses of around 33 billion Euros (Kron et al., 2022; Munich Re, 2022), 20 billion Euros of
these losses in Rhineland-Palatinate (DKKV, 2022). Around 42,000 inhabitants were affected
by the flood along the Ahr and around 8,800 buildings were damaged (DKKV, 2022). More
than 475 buildings were completely destroyed or had to be demolished due to the severity of
the damage, including 200 residential buildings (Kron et al., 2022).
The last significant flood event on the Ahr before 2021 was in June 2016, but with a water level
of 3.71 metres at the Altenahr gauge and no fatalities, the flood was significantly less extreme
than that of 2021 (Landesamt für Umwelt Rheinland-Pfalz, 2016). A flood of a similar
magnitude to the one in 2021 occurred at the Ahr in 1804 (Roggenkamp & Hergert, 2022). The
reconstructed discharge at the gauge Altenahr for 1804 was 1090 m³/s, which is assumed to be
similar to the peak discharge for the July 2021 flood, peak water level in 1804 was estimated
to have been 7.29 meters (Vorogushyn et al., 2022). Reconstructions for the Altenahr gauge
suggest a maximum water depth of 10.2 meters in July 2021 (Apel et al., 2022).
The objective of this study is a particularly detailed analysis of the flood fatalities in the Ahr
valley in 2021 based on the death investigation files of the public prosecutor's office. The
analysis is structured according to the risk concept into hazard, exposure and vulnerability
factors (United Nations Office for Disaster Risk Reduction, 2015). The analysis intends to
improve the knowledge on the causes of flood fatalities in order to support flood risk
management and prevent future fatalities.

## 2. Data and Methods
The data basis for this analysis are the death investigation files from the public prosecutor's
office in Koblenz, Rhineland-Palatinate, Germany. One file per flood fatality in the Ahr valley
in July 2021 was analysed, i.e., 134 files. The files include a description of the victims including
age, gender, sometimes health aspects, etc. In many cases, they contain conversation transcripts
with witnesses who describe the course of the accident and provide details about the



circumstances of the death. No autopsies were performed for the flood fatalities and death by
drowning was presumed, making it impossible to analyse the medical cause of death. Further,
the accident location and location of discovery are included in the police reports, as well as the
time and day of the discovery of the body.
The data was anonymised and classified according to the coding system of Thieken et al.
(2023a). The coding system covers the following aspects: gender, age, nationality, mobility
and cognitive impairment, location of accident, location of discovery, time of day, temporal
relationship to event, personal relationship to location, activity, accident dynamics and medical
cause of death. For quality control, the coding was carried out independently by two people.
The coding results were compared and, in the event of discrepancies, the information in the
files was checked again in detail, its interpretation discussed, and the most likely class assigned.
Accident locations provided in the files were geocoded. In 19% of the cases, only the place of
discovery is known. In these cases, the place of discovery was geocoded. In these cases, it is
not clear if the accident had occurred outside or indoors and how close the place of discovery
was to the location of the accident.
These locations were overlaid with the official hazard map for an extreme event available in
July 2021 (Ministerium für Klimaschutz, Umwelt, Energie und Mobilität Rheinland-Pfalz,
2021) as well as with the reconstructed flood maps by Apel et al. (2022) that provide flow
velocity and water depth, using QGIS. For the indoor accident locations, the maximum water
depth and flow velocity directly outside the building was recorded.
**3. Results and Discussion**
**3.1 Hazard factors**
The probability of fatal accidents during floods increases with more severe hazard impacts.
The simulated maximum water depth and flow velocities at the accident locations were high,
with 73% of the 109 analysed cases experiencing more than 2 m of water depth and 32% of
cases more than 1 m/s flow velocity (Figure 3). Both, water depth and flow velocity were
extreme, with more than 10 meters of water depth estimated at the Altenahr gauge (Apel et al.,
2022). The hazard pattern differs between the indoor and outdoor locations, as flow velocity
outside the building may not have played a role in accidents that happened inside the building.
Even relatively shallow water depths on the outside can lead to fatal accidents in cellars if the
water enters the building. However, many indoor accident locations had particularly high water
levels of more than 4 metres (32% of all indoor cases), so that fatalities occurred on the ground
floor and upper floors (Figure 3).
Accident locations outdoors were more often associated with high flow velocity, with the
maximum at 4.5 m/s (Figure 3). The hazard pattern for outdoor cases shows that people died
at shallower water depths where the flow velocity was high. The combined hazard impact of
water level multiplied by flow velocity is decisive for destabilising people standing in the flood
water. The critical value for human instability is estimated at 1 m²/s (Jonkman and Penning-
Rowsell, 2008, Apel et al., 2022).



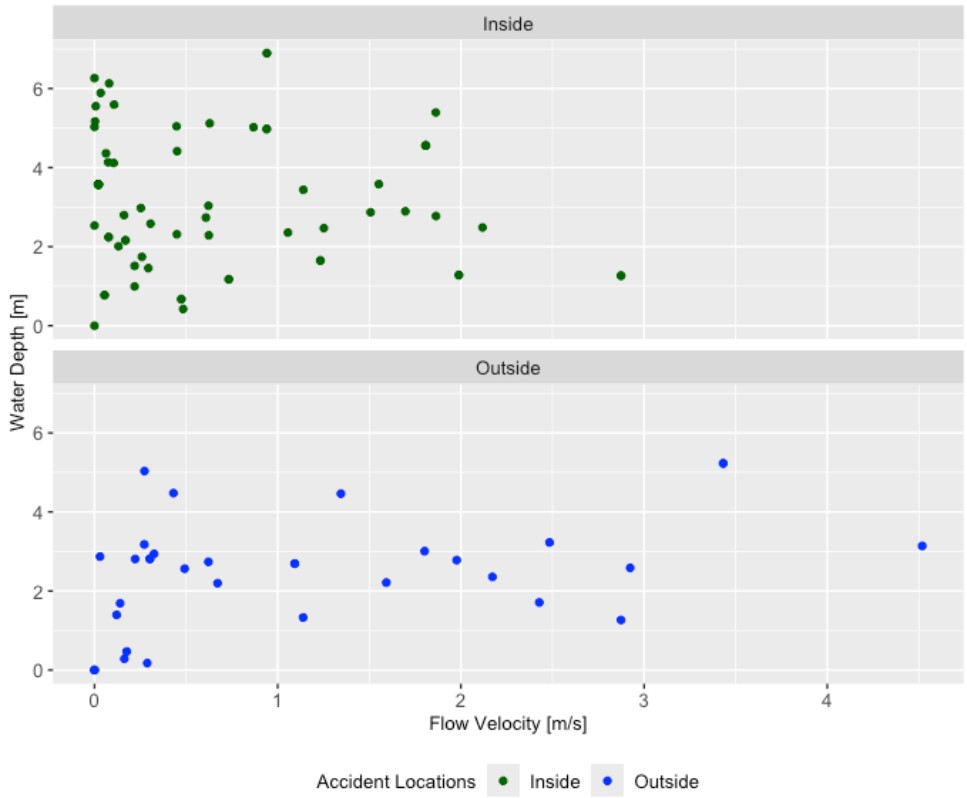

Figure 3: Scatterplots of reconstructed water depth and flow velocity (Apel et al., 2022) at inside and outside locations. The analysis excludes one accident location (6 fatalities) upstream in Dorsel, which is not covered by the reconstructed water depth and flow velocity maps (Apel et al., 2022) and cases where the accident location is unknown (19 fatalities).

## 3.2 Exposure factors

The accident locations analysed in relation to the official hazard map for the extreme scenario (Ministerium für Klimaschutz, Umwelt, Energie und Mobilität Rheinland-Pfalz, 2018) reveal that 75% of the fatalities occurred outside of the mapped hazard zones (Figure 4). The inundation extent and water depth of the unprecedented flash flood in 2021 far exceeded the extreme scenario of the official hazard maps (Figure 2). Considering that official hazard maps are used to decide on evacuations and emergency response, the inaccuracy of the maps may have led to sub-optimal decisions (Kron et al., 2022). They probably gave a false sense of security for areas outside the mapped extreme flood scenario. We therefore recommend improving the development and presentation of worst-case scenarios for official hazard maps and expanding the use of impact forecasting, as it can significantly improve emergency management of unprecedented floods (Apel et al., 2022, Merz et al., 2024).



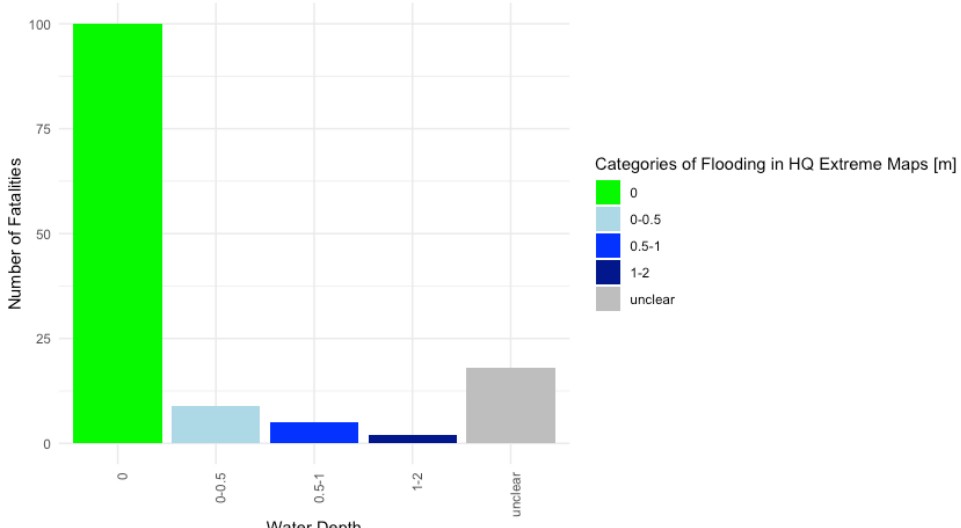

Figure 4: Accident locations in relation to the HQ Extreme Maps (Ministerium für Umwelt,
Energie und Mobilität Rheinland-Pfalz, 2018) categorized in classes of meters of flooding.

Most of the fatal accidents, 65%, occurred indoors, which is probably related to the fact that
the flood peak was reached between 1 and 2 am in Bad Neuenahr-Ahrweiler and Sinzig, where
most of the fatalities occurred (Mohr et al., 2022). The lack of warning and timely evacuation
played a role as well (Thieken et al., 2023b). Of all indoor accidents, 11% happened in cellars
and 3% in basement apartments (Figure 4), particularly dangerous locations during flooding.
Cellars can become traps, as even the pressure of small amounts of water can make it
impossible to open the cellar door again. Flash flood emergency communication should clearly
recommend not going into the cellar to check the heating or safe belongings, which is suggested
before slowly rising river floods with sufficient time for emergency action (Kreibich et al.,
2021). Basement apartments can make it difficult for the residents to take refuge on higher
floors. However, with 37% most indoor accidents occurred at the ground floor and some even
at higher floors (5%), locations which are commonly not perceived as being particularly
dangerous (Figure 4).
The campsite location in Dorsel was the first accident location along the Ahr where fatalities
occurred, even when the flood had not reached its peak yet. According to newspaper reports
the campsite flooded at around 4 pm on July 14th without the residents having received warning
or evacuation messages (FOCUS online, 2022). They were highly exposed as their mobile
homes offered no protection from the floods. Campsites are generally considered dangerous
places during floods, as people are not only highly exposed, but are often non-residents, less
aware of local conditions and news, and more difficult to reach with warning (Terti et al., 2017,
Aceto et al., 2017). However, with 18% most outside accidents occurred just on a street (Figure
5), a place that is not expected to be particularly dangerous, unlike places near a river or a
bridge.



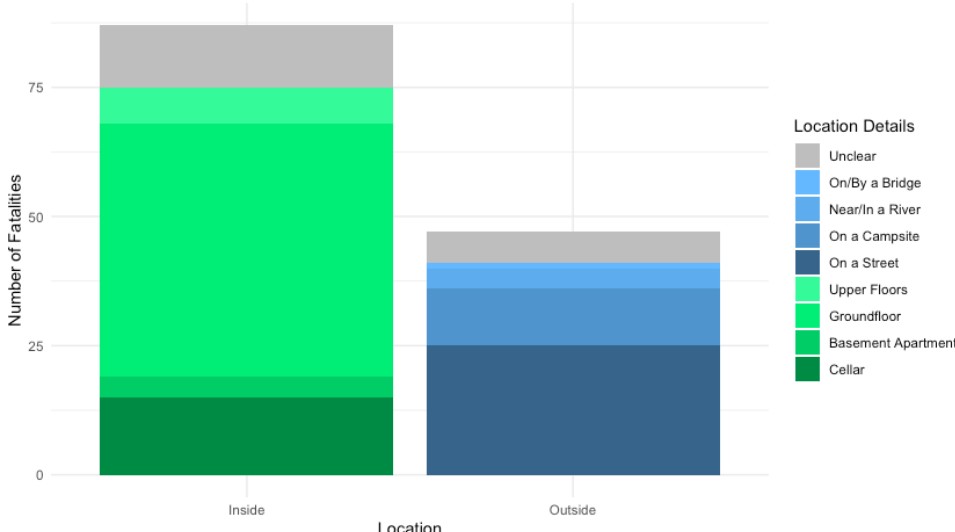

Figure 5: Location of fatalities (inside and outside).

### 3.3 Vulnerability factors

During the food in the Ahr valley, the elderly were particularly vulnerable. 80% of the victims were aged 60 or over (Figure 6). In contrast, most flood victims in Europe between 1980 and 2018 were between 30 and 64 years old (Petrucci, 2022). Compared to the total population of the federal state of Rhineland-Palatinate, Germany, the proportion of elderly people who died during the floods is significantly higher than the proportion of the total population. This high vulnerability of the elderly might be due to their physical limitations and difficulties in moving to higher stories. Petrucci et al. (2019) showed that fatal accidents with older people commonly occur indoors at home. Men over the age of 70 appear to be particularly vulnerable due to their high susceptibility to trauma (Kellar and Schmidlin, 2012). However, no gender-specific differences in vulnerability were observed. 49% of the victims were male and 51% were female (Figure 6), which matches the gender distribution of the total population of Rhineland-Palatinate in December 2020 (Statistisches Landesamt Rheinland-Pfalz, 2020). This is consistent with previous findings of balanced gender distributions of fatalities in flash floods, where most victims were surprised by the floods (Petrucci, 2022). In other flood situations, previous findings show that a higher proportion of men die, as vehicle-related accidents and risky behaviour, including rescue operations, are more likely to play a role (Sharif et al., 2012).

16% of the flood victims had a disability. There were records of mobility impairments for 7 victims, while 14 were recorded as having cognitive impairments. This high number is due to fatalities that occurred in a residential home for adults with mental disabilities. There are relatively few studies that investigate disabilities of flood victims, however, there is one report from Italy in 2000, where 13 people with mobility impairments died at a campsite during the flood (Aceto et al., 2017). Thus, it is important to pay particular attention to these groups during evacuations, e.g., by giving special attention to hospitals, retirement homes and homes for the disabled.



257

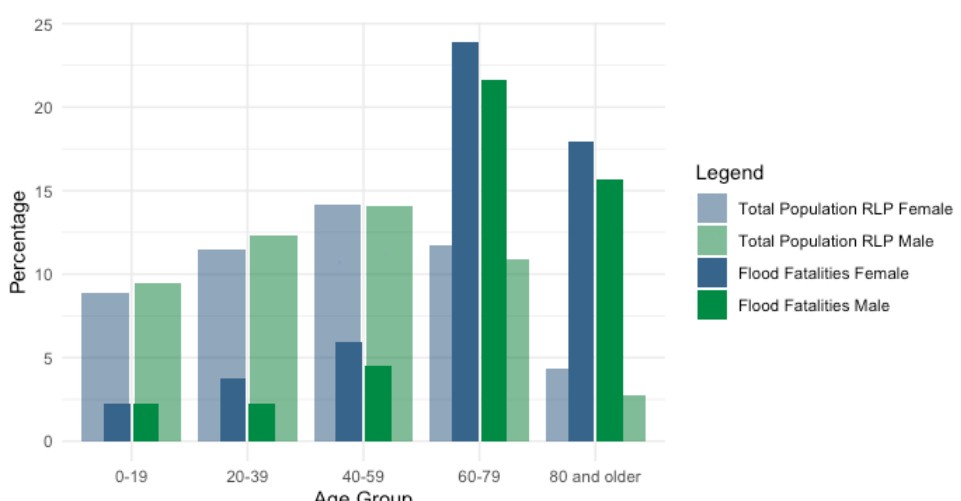

258

Figure 6: Age distribution by gender for the July 2021 flood fatalities and the total population
of Rhineland-Palatinate (Statistisches Landesamt Rheinland-Pfalz, 2020).

261

**Conclusions**

To minimise the number of fatalities from flooding, our recommendation is to improve risk
management of unprecedented flash floods (Kreibich et al., 2022), as 75% of the fatalities
occurred outside of the officially mapped hazard zones. The development of worst-case
scenarios needs to be improved, including better presentation of extreme events in hazard maps
(Merz et al., 2024), so that decision-makers and the public can better prepare for such extreme
events. Additionally, impact forecasting can significantly improve emergency management of
unprecedented floods (Apel et al., 2022). Public understanding of the particular risk of extreme
flash floods must be improved through risk communication, in particular by raising awareness
of dangerous locations, behaviours and vulnerable groups. Campsites, cellars and basement
flats are identified as particularly dangerous places during floods (Terti et al., 2017, Aceto et
al., 2017, Papagiannaki et al., 2022). However, during the 2021 flood many have also died on
the ground floor and in the street, places that are not normally considered particularly
dangerous. Thus, in the specific case of an extreme flash flood, the focus of emergency
communication needs to be turned away from mitigating economic damage to saving human
lives. Warning messages must clearly communicate that those at risk should move to a safe
place and when it may be too late to leave the building to go to an upper floor. Elderly people
and people with cognitive or mobility impairments are particularly vulnerable. It is therefore
important to pay particular attention to these groups during evacuations, e.g., by giving special
attention to hospitals, retirement homes and homes for the disabled.

282

**Acknowledgments**

We would like to thank the public prosecutor's office for allowing us to analyse the death
investigation files related to the 2021 flood in anonymised form. We thank Rumyana Zimmer
and Astrid Krahn for their work on data coding and quality control. We thank the German



Federal Ministry of Education and Research (BMBF) for financial support within the
framework of the KAHR and AVOSS projects (grant no. FKZ 01LR2102F, grant no. FKZ
02WEE1629C).

**Code/Data Availability**
Parts of the anonymised flood fatality data may be obtained upon request.

**Author Contributions**
BR: Conceptualization, Data curation, Formal Analysis, Supervision, Visualization, Writing –
original draft, Writing – review & editing, HK: Conceptualization, Writing – review & editing,
Supervision

**Competing Interests**
HK is member of the editorial board of Natural Hazards and Earth System Sciences.

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
