# Peer review of "Causes of the exceptionally high number of fatalities in the Ahr valley, Germany, during the 2021 flood"

_EGUsphere, 2024_

## Referee Comment (RC2)

**Causes of the exceptionally high number of fatalities in the Ahr valley, Germany, during the 2021 flood**

*General comments*

The paper explains the factors which made the flood event the deadliest one in Germany in more than 40 years by studying the circumstances and backgrounds of the individual fatality reports. The paper is well written, easy to read and clear.

The most important factors that contributed to the large number of fatalities in this event are:

- The extreme severity of the flooding with both high water levels and extremely high flow velocities.
- The underestimation of the hazard and its potential consequences:
    - The underestimated flood forecasts 24 hours before the flooding,
    - The lack of awareness at the governments and public of the possibility of these severe types of floods in this area and the potential to be life threatning.
- The ineffective warning: the warning which did not reach everybody and did not convey clearly the message that a life-threatning event might occur and that people needed to find a safe location.
- The incapacity to rescue oneself of elderly and people with a disability.

Some minor suggestions are provided below to enhance the discussion on the analysis of the factors contributing to the fatality numbers or to enhance the clarity of the paper even further.

*Specific comments*

Abstract: The abstract is clearly written. It describes the events and demonstrates its extremity, it provides information on where people died and provides recommendations to reduce the probability of such disasters in the future. However, it does not summarize the causes of the high number of fatalities as is promised in the title. This might be added in a summarized form.

Introduction: This shows the exceptional character of the event in number of fatalities. The text in lines 75-80 may be removed. It does not add to the explanation of why the Ahr flood resulted in that many fatalities.

In the text and abstract the flood is sometimes referred to as unprecedented flood, but it is also explained that a similar event occurred in 1804. Perhaps one sentence can be added explaining this: the flood is called unprecedented, but this refers to the recent history. There was a similar flooding in 1804, but then the valley was still completely different and also the village was different and most people are not aware of this past flood anymore.

Section 2: In the text it is repeated three times that data is used from files from the public prosecutor's office In Koblenz. T(his is not a big issue, but perhaps it may be repeated less).

Section 3: results and discussion: Add a brief sentence explaining in one sentence why 109 cases are analysed. It is because of 134 there are 25 which are out of the area or without information. Cases refer to fatality incidents.

Add some context in section 3 if possible: The analysis focused of corse on the fatality records. To get a complete insight into the occurrence of flood fatalities however, also locations without fatalities should be considered. This may be mentioned without adding such an analysis to this paper. The paper says that in more than 73% of the cases (flood fatalities) water depths exceeded 2m. Is there also info on e.g. how many residences experienced these conditions in total (with and without fatalities). Thus: did you also consider locations without fatalities with similar conditions and looked at how people there survived? Might that add to the understanding of why the other people died? What is the mortality of the event in total? Are the fatalities rare exceptions/incidents or did people at certain very dangerous locations had a seriously higher mortality than at other locations?

Line 174: you mention cellars as dangerous, but you indicate that water depths were above 4 meters so that many fatalities occurred on the ground floor. Do you know if many people died in cellars? If not, perhaps mention that. It is a little confusing now.

Line 239-240 "Men over the age of 70 .. , 2012)"this sentence may be removed. It is not relevant here.

Line 275: This line suggests that the focus of emergency communication was on mitigating economic damage before. This was not explained before. What was the message? Give that more attention in the paper, since that may also be a contributing factor to why that many fatalities occurred.

---

## Author Response (AR1)

**Response letter for egusphere-2024-2066 - Causes of the exceptionally high number of fatalities in the Ahr valley, Germany, during the 2021 flood by Belinda Rhein and Heidi Kreibich**

Dear Editor Animesh Gain, Dear Reviewes Katerina Papagiannaki and Karin M. de Bruijn,
Thank you very much for your thoughtful and constructive feedbacks on our manuscript entitled "Causes of the exceptionally high number of fatalities in the Ahr valley, Germany, during the 2021 flood". Please find our point-by-point response below; our responses are marked with "R:".
Best regards
Belinda Rhein and Heidi Kreibich

**Response to reviewer Katerina Papagiannaki (reviewer #1)**

**General comments**

The aim of this paper fits perfectly with the aim and scope of the journal. It is about one of the most catastrophic floods in recent history in Europe, and its analysis is of high interest from many perspectives. The study of fatalities, circumstances and causes is critical. The article is very well-written and presents the results and recommendations clearly and concisely. I believe it merits being accepted for publication, and I only have some minor comments below.

R: Thank you very much for the positive feedback to our study

**Abstract**

L39: 'Specific recommendations are that in the event of such extreme flash floods, the warning messages must clearly communicate that saving human lives must be the priority,..'

Qu.: To whom will this message (that saving human lives must be the priority) be delivered? It sounds like a message to managers, not to the people. However, the specification in the same sentence (those at risk should move to safe places, e.g., to the upper floors') sounds like targeting people. I think this should be a bit modified, as I assume that a message to the people would not include that saving human lives must be the priority.

R: We clarify to whom the recommendation is directed, namely to disaster risk manager. Thus, we reformulate as follows: "Specific recommendations for disaster management are that in the event of such extreme flash floods, the warning messages must focus on saving human lives, i.e., those at risk must be advised to move to safe places, e.g., to the upper floors instead of trying to save belongings."

**Data and Methods**

153: 'In 19% of the cases, only the place of discovery is known''

Qu.: Are these excluded from the analyses? Based on the results, yes; thus, I recommend clarifying this here.

R: We clarify that the cases with unknown accident location were excluded from the analysis. We reformulated the sentences as follows: "In 19% of the cases, only the place of discovery is known; these cases were excluded from the analysis. Accident locations were overlaid with the official hazard map for an extreme event available in July 2021 (Ministerium für Klimaschutz, Umwelt, Energie und Mobilität Rheinland-Pfalz, 2021) as well as with the reconstructed flood maps by Apel et al. (2022) that provide flow velocity and water depth, using QGIS."

154-155: 'In these cases, the place of discovery was geocoded. In these cases, it is not clear if the accident had occurred outside or indoors and how close the place of discovery was to the location of the accident'

Qu: I suggest rephrasing or merging the 2 sentences, as to repeat the phrase 'In these cases' is not so nice.

How many are the uncertain location data?

R: We clarify that only the cases with known accident location are used for the analysis, the sentences on lines 154-155 (mentioned above) are deleted.

 **Results and Discussion**

165: 'The probability of fatal accidents during floods increases with more severe hazard impacts'

178: 'The combined hazard impact of water level multiplied by flow velocity is decisive for destabilising people standing in the flood water.'

Qu: I suppose the authors mean 'hazard severity' instead of 'hazard impact'. Even if the velocity and depth are consequences of the rainfall hazard, they are still considered associated hazards. The word 'impact' is a bit confusing, especially when the analysis focuses on flood-induced impacts, i.e. human fatalities.

R: We exchange "impact" with severity. Thus, sentences read as follows: "The probability of fatal accidents during floods increases with more severe hazard." and "The combined hazard severity of water level multiplied by flow velocity is decisive for destabilising people standing in the flood water."

196: 'We therefore recommend improving the development and presentation of worst-case scenarios for official hazard maps and expanding the use of impact forecasting, as it can significantly improve emergency management of unprecedented floods'

Qu: Based on the previous comment, the impact forecasting here is unclear. Does it concern fatalities or water depth and velocities?

R: We understand impact forecasting in a broad sense, including both, water depth but also damage hotspots. Thus, we reformulated the sentence: "We therefore recommend improving the development and presentation of worst-case scenarios for official hazard maps (Merz et al., 2024). Additionally, expanding the use of impact forecasting, i.e. warnings of inundation (water depth, flow velocity), blocked roads or damage hotspots, can significantly improve emergency management of unprecedented floods (Rözer et al. 2021, Apel et al., 2022)."

**Response to reviewer Karin M. de Bruijn (reviewer #2)**

**General comments**

The paper explains the factors which made the flood event the deadliest one in Germany in more than 40 years by studying the circumstances and backgrounds of the individual fatality reports. The paper is well written, easy to read and clear.

The most important factors that contributed to the large number of fatalities in this event are:

- The extreme severity of the flooding with both high water levels and extremely high flow velocities.
- The underestimation of the hazard and its potential consequences:
  - The underestimated flood forecasts 24 hours before the flooding,
  - The lack of awareness at the governments and public of the possibility of these severe types of floods in this area and the potential to be life threatening.
- The ineffective warning: the warning which did not reach everybody and did not convey clearly the message that a life-threatning event might occur and that people needed to find a safe location.
- The incapacity to rescue oneself of elderly and people with a disability.

R: Thank you very much for the positive feedback to our study

Some minor suggestions are provided below to enhance the discussion on the analysis of the factors contributing to the fatality numbers or to enhance the clarity of the paper even further.

**Specific comments/minor suggestions**

Abstract: The abstract is clearly written. It describes the events and demonstrates its extremity, it provides information on where people died and provides recommendations to reduce the probability of such disasters in the future. However, it does not summarize the

causes of the high number of fatalities as is promised in the title. This might be added in a summarized form.

R: We add the following sentence to the Abstract: "Thus, the main causes for the exceptionally high number of fatalities were the extreme severity of the flood and its underestimation by the population and authorities, as well as inadequate early warning and evacuation."

In the text and abstract the flood is sometimes referred to as unprecedented flood, but it is also explained that a similar event occurred in 1804. Perhaps one sentence can be added explaining this: the flood is called unprecedented, but this refers to the recent history. There was a similar flooding in 1804, but then the valley was still completely different and also the village was different and most people are not aware of this past flood anymore.

R: We clarify our understanding of "unprecedented" and add the following sentences: "The flood of July 2021 can be described as an unprecedented flash flood (Kreibich et al., 2022) with a particularly high number of deaths, especially with 134 fatalities in the Ahr valley in Germany (Koks et al., 2021). The term 'unprecedented' is used here in a subjective sense, i.e. local residents and authorities have never experienced a flood with a similar severity and number of fatalities." and "A flood of a similar magnitude to the one in 2021 occurred at the Ahr in 1804 (Roggenkamp & Hergert, 2022). However, this event was too long ago for the local population and authorities to actively remember it."

Add some context in section 3 if possible: The analysis focused of course on the fatality records. To get a complete insight into the occurrence of flood fatalities however, also locations without fatalities should be considered. This may be mentioned without adding such an analysis to this paper. The paper says that in more than 73% of the cases (flood fatalities) water depths exceeded 2m. Is there also info on e.g. how many residences experienced these conditions in total (with and without fatalities). Thus: did you also consider locations without fatalities with similar conditions and looked at how people there survived? Might that add to the understanding of why the other people died? What is the mortality of the event in total? Are the fatalities rare exceptions/incidents or did people at certain very dangerous locations had a seriously higher mortality than at other locations?

R: Indeed, we focus our analysis on the detailed information provided in the fatality records, other analyses are out of scope of this study. We add the following discussion: "In addition to the severity of the hazard, other factors such as exposure and vulnerability characteristics can also influence fatal accidents; corresponding results based on analyses of the fatality records are presented in the following sections. To gain further information from a different perspective, it would also be interesting to analyse control groups, i.e. to compare why deaths occurred in one situation and not in a comparable situation. For example, one could analyse hazard hotspots with and without fatalities to determine how people survive in extremely dangerous conditions. However, this approach is out of scope of our study."

Line 174: you mention cellars as dangerous, but you indicate that water depths were above 4 meters so that many fatalities occurred on the ground floor. Do you know if many people died in cellars? If not, perhaps mention that. It is a little confusing now.

R: We provide the information on indoor locations in sub-section 3.2. However, for better understanding of water depth and locations, we add the following sentence on line 182: "Of all indoor accidents, 11% happened in cellars, 37% at the ground floor and 5% at higher floors (Figure 5)."

Line 275: This line suggests that the focus of emergency communication was on mitigating economic damage before. This was not explained before. What was the message? Give that more attention in the paper, since that may also be a contributing factor to why that many fatalities occurred.

R: In the event of slowly rising river floods, those affected can be advised to limit their damage by saving their belongings, e.g. by driving their cars to flood-free areas. However, to the best of our knowledge, no such advice was given during the 2021 floods. To avoid misunderstanding, we change the sentence as follows: "Thus, in the specific case of an extreme flash flood, the focus of emergency communication needs to be on saving human lives."